# MOTH: Moving Object Tracking via Head-Mounted Displays

Fabian Bösing
fabian_boesing@arcor.com
Karlsruhe Institute of Technology
Karlsruhe, Germany

David Puljiz
david.puljiz@h_ka.de
Karlsruhe University of Applied
Sciences
Karlsruhe, Germany

Björn Hein
bjoern.hein@h_ka.de
Karlsruhe University of Applied
Sciences
Karlsruhe, Germany

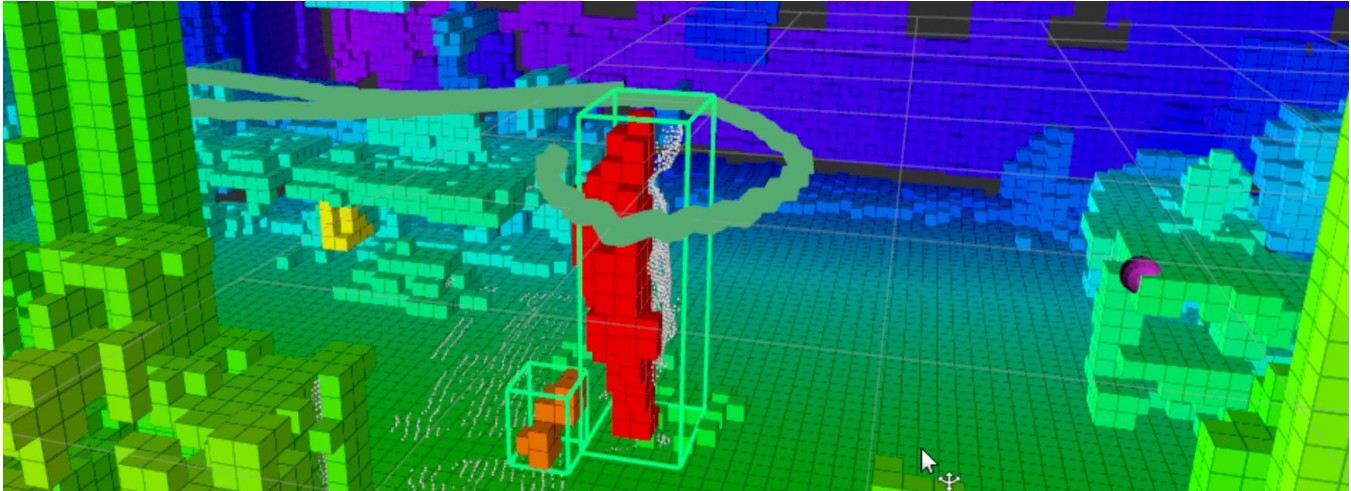

Figure 1: Example of the voxel based moving object tracking approach used to track a person in an indoor setting.

## ABSTRACT

Modern head-mounted displays (HMDs) carry a plethora of sensors, comparable to autonomous systems such as robots or self-driving cars. We can thus leverage algorithms and methods developed for those systems to support and help users wearing HMDs. Using the HoloLens 2 as our system of choice, we explore different methods for tracking moving objects. An exhaustive review of current methods and their applicability to HMDs is conducted. Two methods are selected as basis for implementation - a voxel grid based and cluster based approach. We modify the two approaches to be suitable for use on HMDs. After qualitative analysis, they were compared to a ground truth obtained via a marker tracking system. The overall accuracy of tracking moving humans was shown to be between 15 and 25 cm. Though the primary motivation was to facilitate human-robot collaboration in indoor industrial facilities via augmented reality glasses, the research can easily be applied to any Mixed Reality (MR) device possessing appropriate sensors. Likewise, such systems may be of use outdoors as MR slowly exits into the open.

*VAM-HRI '23, March 13, 2023, Stockholm, Sweden*
© 2023 Association for Computing Machinery.
ACM ISBN 978-x-xxxx-xxxx-x/YY/MM. . . $15.00
https://doi.org/

## CCS CONCEPTS

• **Human-centered computing → Mixed / augmented reality**; *Mobile devices*; • **Computer systems organization → External interfaces for robotics**; • **Computing methodologies → Tracking**.

## KEYWORDS

Moving Object Tracking, Augmented Reality, Head-Mounted Displays

**ACM Reference Format:**
Fabian Bösing, David Puljiz, and Björn Hein. 2023. MOTH: Moving Object Tracking via Head-Mounted Displays. In *Proceedings of Make sure to enter the correct conference title from your rights confirmation emai (VAM-HRI '23)*. ACM, New York, NY, USA, 9 pages. https://doi.org/

## 1 INTRODUCTION

Moving object detection (MOD) and tracking (MOT) are important tasks in the contexts of both human-robot interaction and autonomous driving. Therefore, there exist numerous approaches for tackling those tasks. So far, virtually all of these approaches use sensors which are either mounted on a mobile robot[5, 8, 43], placed in fixed locations in the environment[26, 27, 33, 41] or mounted on a car[4, 14, 19]. Though MOT is paramount for safety in both robotics and self-driving cars, there are also contexts in which it may be useful to implement it on wearables such as HMDs, creating MOTH. In the context of human-robot interaction, tracking of moving robot arms or mobile robots ensure situational awareness

and an additional level of safety. Tracking data may also be used for collaborative localisation and referencing, both between robots and HMDs as well as between HMDs themselves.

Though the main motivation for this work was Human-Robot Interaction (HRI) in an industrial setting, MOTH may also be of use in recreation e.g. warning bicycle drivers of possible collisions or tracking trajectories of a basketball to improve aim.

MOT is well researched in the context of autonomous systems. This large body of research is used as basis for developing MOTH. The main difference is that autonomous system do not have as much weight or computational restrictions as HMDs. Use of relatively heavy LiDARs and 360° cameras, as well as computationally expensive deep learning approaches is quite common. Thus a detailed review of MOT approaches for autonomous systems is needed as well as a thorough analysis of their applicability on HMDs.

The main contributions of the present work are:

- Detailed analysis of applicability of MOT approaches developed for autonomous systems for HMDs
- Adapting selected approaches for use on HMDs
- Qualitative and quantitative tests of the developed algorithms

## 2 STATE OF THE ART

MOD refers to the segmentation of non-stationary objects of interest from the surrounding area or region [23] in a sequence of data, captured using one or multiple sensors, such as monocular and stereo cameras, time-of-flight (ToF) or LiDAR sensors, Radar etc. [2]. Tracking dynamic object over time is the task of MOT. These two tasks are related as MOT uses the results of MOD as its input in most cases. MOD approaches may work in 2D space (images), 3D space (range data), or a combination of both [2].

### 2.1 Image based approaches

Klappstein *et al.* [22] proposed a MOD algorithm which computes the optical flow and disparity map of an RGB stereo pair. The approach reconstructs the 3D position of each pixel and assigns it a velocity. The egomotion of the camera can be calculated, and points deviating from the egomotion - moving objects - can be segmented out using a graph-cut algorithm. The paper lacks a quantitative evaluation which makes comparison to other approaches difficult.

In [20],MOT for a stationary monocular camera was presented. The optical flow is estimated on a greyscale image. After applying a median filter for outlier removal, moving objects are detected via a thresholding function. For each segmented moving object a motion vector is calculated. The authors reported an accuracy of 90%.

Lin *et al.* [24] use a disparity map converted to a depth image. The ground plane is detected, and objects above the plane are segmented out. For each object, SURF [6] features are extracted and matched against the features from the previous frame. The camera motion is estimated using visual odometry. SURF feature matching is used to distinguish between moving and static objects using an adaptive threshold. Tests using three video sequences found 745 and 96 misclassifications for moving and static objects respectively. However, neither the amount of correct classifications nor the amount of total objects was given.

Popović *et al.* [34] proposed an approach that tracks the disparity of each pixel using Kalman filters. By comparing the calculated and predicted disparity maps, pixels corresponding to potential moving objects are detected. Initial detections are refined by searching for areas with large differences using a sliding window approach. The detected areas are iteratively merged using a greedy algorithm. The authors compared their algorithm against OpenCV's Semi-Global Mapping (SGM) implementation and found that their proposed algorithm "had on average fewer outliers in 6 out of 7 sequences"[34] while also being faster to compute.

Recent MOD and MOT approaches rely mostly on neural networks. In [1] a 2D object tracking framework for usage in AR on smartphones was presented. ORB [40] features and optical flow are used to track moving objects, while the neural network is used only to detect new objects or objects which are not reliably tracked by the classical tracker. On a test set of 80 videos, the average classification precision was slightly above 60%, with an average Intersection Over Union (IOU) of slightly below 40%, both similar to the baseline of always using Tiny YOLO for object detection, while requiring on average 34.5% less power.

Ramzy *et al.*[38] presented two neural network architectures for MOD. One uses the RGB image as input and the other the optical flow. The outputs are fused using either Long short-term memory (LSTM) or a gated recurrent unit (GRU), while a decoder finally applies a pixel mask of moving objects on the input image. Experimental evaluation on the KITTI-Motion dataset found that the two network architectures achieved a mean IOU of 82.5% (GRU) and 83.7% (LTSM) while being able to process 23 and 21 frames per second respectively on a Titan X Pascal GPU.[38]

### 2.2 Range sensor based approaches

The first group relies on voxelization of the captured point cloud data, such as [3, 4, 8, 15, 32]. While some of these approaches use the voxelized representation to detect moving objects based on voxels whose occupancy changes between consecutive frames[4, 8], other approaches first build an occupancy map of all static voxels which is then used as a binary filter to remove all points belonging to moving objects[32]. Alternatively, such a voxelized representation may also be used for segmentation of the captured point cloud into individual objects which are then processed further[15].

Azim *et al.*[4] compare the voxelized representation of current and previous frames on a per-voxel-basis, each voxel being classified either static or dynamic. Static voxels are used to generate a map of the static environment, whereas dynamic voxels are clustered to detect moving objects. Objects are classified based on their size, before being tracked using a Kalman filter with a modified GNN data association scheme. The approach was evaluated with a vehicle-mounted Velodyne HDL-64 LIDAR in urban traffic scenarios. It performed adequately despite considerable amount of noise, though there was no ground-truth comparison.

In [8] a preprocessing step is applied to the captured point cloud by detecting and removing the ground plane, followed by a down-sampling step before generating the voxel grid. The tracking and detection are performed using a Kalman filter with joint probabilistic data association. The approach was tested in an indoor and an outdoor scene with a Velodyne HDL-32E LiDAR mounted to a

mobile platform. Qualitative analysis provided adequate results at 5 FPS, though, again, ground truth was unavailable.

Other approaches cluster the point cloud objects after the ground plane is removed [11, 17, 29, 44–46]. The clusters can then be processed further to distinguish between static and moving objects.

In [17] data from a geospatial map, such as OpenStreetMap, is used to remove points corresponding to objects outside the road, such as buildings. The remaining points are clustered based on their Euclidean distance, before being compared to the previous frame. The motion vector for each match is calculated, based on which the object is classified as being static or dynamic. The latter are tracked using a Kalman filter. In an experimental evaluation based on a driving sequence of the KITTI dataset, the approach tracked a cyclist with an error below 0.25 meters. However, the authors also found that some static objects were falsely classified as dynamic.

The approach by Yan *et al.* [45, 46] removes the ground plane using a threshold filter on the points' z-value. The remaining points are clustered with an adaptive euclidean distance threshold. An unscented Kalman Filter is used for tracking while a Support Vector Machine (SVM) classifies the clusters as humans or other objects. The SVM can be retrained online. The clustering algorithm's runtime is 74.4 milliseconds per frame on an Intel Core i7 7560U CPU. An online trained classifier achieved an average precision of 59.8% with a stationary mobile robot and 41.2% with a mobile one.

In [19], 3D flow field analysis is used to detect moving objects. Points belonging to a smooth flow between two consecutive point clouds belong to the static environment, while clusters of points with sparse flow are moving objects. An Intel Core i7 CPU took 5.57 seconds to process 87 frames using the proposed algorithm, resulting in 15 Fps. The moving object detector has a recall of 90.1% and a true negative rate of 98.5%.

Recent approaches, such as [7, 12, 30, 31, 47], tend to use neural networks. Nakamura *et al.* [30] proposed a technique to avoid momentary missed detections by using a Point Pillar network, which analyses a 2D array of pillars, instead of using a voxel-based approach. The network outputs the 3D bounding boxes and the classes of the detected objects. The approach was able to process between 12.10 and 16.84 FPS, with an average precision between 29.42% and 89.83%, depending on the exact implementation.

Fang *et al.* [12] present a single object tracking approach using a neural network with two sub-nets working on raw point cloud data. The approach was able to process 20.8 frames per second on a Nvidia GTX 1080Ti, tracking a single car in the KITTI dataset with a 3D success of 57.25%, a 3D precision of 75.03%, a bird's-eye-view success of 73.02% and a bird's-eye-view precision of 79.45%.

## 2.3 Fusion based approaches

Fusion based approaches fuse 2D image data and range sensor data. Approaches such as [9, 10, 18, 21] accomplish this by detecting possible moving objects for each type of sensor separately. The detections are then fused and validated, before being tracked.

Chavez-Garcia *et al.* [9] employed two object detectors for data obtained by a RADAR and a LiDAR sensor respectively, while a camera was used to determine object classes. The classifications are fused using Bayesian based fusion and Evidental Theory, and tracked using a model-based MOT approach. The approach was

found to have an average processing time of 40 milliseconds per frame in urban areas and 30 milliseconds per frame on highways and in rural areas. It was able to detect between 83.3% and 100% of all objects correctly depending on object class and test scenario, while misclassifying between 0.1% and 10.8% of all objects.

Approaches presented in [28, 35], use one type of sensor to detect objects and use the other types of sensors to enrich the found detections with additional information.

Neural network based approaches, such as [14, 39], employ neural networks in an almost end-to-end fashion. The data obtained from the camera and the LiDAR sensor is given to the network as input, either in raw form or with some additional preprocessing steps, and the network outputs the found detections.

In [39], the network receives a raw RGB image, the optical flow calculated from that image, and an optical flow calculated from a point cloud obtained by a LiDAR sensor. The network processes the given input data and outputs a 2D motion mask indicating which pixels of the RGB image correspond to moving objects. An IOU of 75.3% was achieved on the KITTI dataset and 71.2% on the Dark-KITTI dataset. Running on a Nvidia Titan X Pascal GPU, the network was able to process 18 frames per second.

## 3 SELECTION OF ALGORITHMS

The three main criteria for selection were processing power requirements, speed and precision. Though information is processed on a desktop PC due to our use-case of interacting with industrial robot arms, the algorithms should also be able to run on board a HoloLens 2. The desktop PC used had a Intel Core i7-4790K (4 Cores, 8 Threads @ 4.0 GHz) with 16 GB RAM and an Nvidia GTX 750 Ti running an Kubuntu 16.04 (64-bit) OS. The processor power is comparable to the HoloLens 2 (Snapdragon 850 - 8 cores @ 2.96 GHz). Approaches that make heavy use of the GPU or require more powerful processors were therefore disregarded. This included basically all neural network approaches as well as sensor fusion-based approaches. Other requirements were near-online tracking capability as well as the best tracking precision possible.

The first selected approach is [4]. This was partly driven due to the use of a voxel based representation which made it suitable to be combined with the AR based robot cell setup proposed in [36]. Though the approach [4] was published almost ten years ago, it does not mean that voxel based approaches are outdated. There exists newer work such as [3], [25] and [47] which also voxelize the environment. However, the former approach did not meet real-time requirements due to only achieving 0.3 frames per second on "a quad core 3.4 GHz processor"[3] and the latter two approaches rely on neural networks for detecting objects in voxels,thus do net meet the hardware constraints stated before.

The second chosen approach was proposed in [45, 46] due to four reasons: *I)* in addition to tracking moving objects, it can distinguish between humans and non-human objects, *II)* it offers a clustering precision of 89.8% and an average human classification precision of 59.8% while *III)* requiring a low processing effort with an average clustering time of 74.4 milliseconds per frame on an Intel Core i7-7560U, and *IV)* it was already tested successfully in indoor environments which corresponds to our original use case. A similar yet newer approach is presented in [44], though it does not

perform object classification. In addition, the exact runtime is not given, but it appeared capable of running online on an Intel Core i5-8265U. Finally, the results are only described qualitatively, with no quantitative evaluation regarding accuracy or precision.

A third approach was considered, selected from the image based approaches, as the two MOT approaches chosen rely solely on information obtained from the HoloLens 2's depth sensor and its self-localization capability. Due to the hardware constraints, recent vision-based approaches were rejected as they rely on deep neural networks. This left four classical vision-based approaches as possible choices [20, 22, 24, 34]. The approach of Kale *et al.* [20] is constrained to tracking objects in 2D image space as it employs only a single camera, whereas the other three approaches benefit from stereo vision for full 3D MOT. The other three approaches are quite similar, yet can not be compared directly with each other due to different evaluation metrics. The approach of Popović *et al.* [34] was chosen as it appears to run comparatively faster than the computationally cheap OpenCV methods, while also being able to deliver a filtered point cloud of the moving objects from the disparity map difference. Unfortunately, it was quickly found that the camera frames of the HoloLens2 were unsynchronised, even though they were both captured at the same time and streamed to the PC inside the same message. Having parallel read threads likewise didn't solve the problem, nor has implementing a 5 bin ring buffer for each of the stereo images. Each element's timestamp was compared to the timestamps of all elements in the other buffer. However, it proved to be impossible to get properly synchronised images using HoloLens2ForCV. Though rejected, it may still be interesting if syncronized camera data is available.

## 4 IMPLEMENTATION

The system consisted of a HoloLens 2 and a desktop PC running the Robot Operating System (ROS) [37] on a Kubuntu 16.04 OS. Using the research mode of the HoloLens 2 and HoloLens2ForCV[1] the raw 2D depth image may be acquired. This image was encoded as a JSON message and streamed to the desktop PC using ROSBridge[2] to facilitate our use-case of human-robot interaction. The depth stream was then converted to a point cloud, which was downsampled and a radius outlier removal filter used before being registered to the global point cloud. Though this approach was adequate with previous work[36] done with the 1st generation HoloLens, the point cloud of the HoloLens 2 proved to be too noisy due to the higher resolution and range. Noisy border pixels were removed from the depth stream and smoothed out with a 3x3 median filter. Euclidean clustering was used to disregard small clusters. To improve speed, the filtering and clustering steps were applied directly in the 2D depth image space instead of the point cloud. The speed increase comes both from the dimensionality reduction, as well as from the fact that, unlike the point cloud, the depth image is an organized construct facilitating nearest-neighbour searching. A 2D clustering algorithm was used to group neighbouring pixels with similar depth value. This proved to remove all the edge cases, such that the radius outlier removal and the median filter we're no longer necessary. The final algorithm is then as follows.

---

[1]https://github.com/microsoft/HoloLens2ForCV
[2]http://wiki.ros.org/rosbridge_suite

**Algorithm 1** Algorithm for obtaining a point cloud from a Hololens 2 depth image

> **Input:** Depth image $depthImage$,
> Pixel undistortion map $pixelDirections$
> **Output:** Point cloud $points$

1: Initialize $points$ as an empty point cloud
2: $clusters \leftarrow$ EUCLIDEANCLUSTERING($depthImage$)
3: **for each** $cluster \in clusters$ **do**
4:    **if** $cluster$ contains more than $clusterSizeThreshold$ points **then**
5:      Initialize $clusterPoints$ as an empty point cloud
6:      **for each** $pixel \in cluster$ **do**
7:        **if** $pixelDirections$ has an undistortion mapping for $pixel$ **then**
8:          $direction \leftarrow$ undistortion mapping stored for $pixel$
9:          $depth \leftarrow$ depth value stored in $depthImage$ at $pixel$
10:         **if** $minThreshold \leq depth \leq maxThreshold$ **then**
11:           $point \leftarrow direction \cdot depth$
12:           Add $point$ to $clusterPoints$
13:      Downsample $clusterPoints$ for a more uniform point density
14:      Add $clusterPoints$ to $points$
15: Transform all points in $points$ from camera space to world space

## 5 VOXEL GRID BASED APPROACH

Though in [4] a Velodyne HDL-64E LiDAR was used, the authors explicitly stated that the approach is applicable to any range sensor, including the time-of-flight (ToF) sensor of the HoloLens 2.

In [4], the point cloud is transformed into a voxel occupancy grid. Each voxel in the environment is marked as either free with some probability $l_{free}$, occupied with probability $l_{occ}$ or unknown. These are assigned using a beam based inverse sensor model, in which beams are projected towards the detected obstacles from the sensor's position. All the voxels that the ray intersects before hitting the obstacle will be marked free.

In this work, the voxel grid was created using OctoMap [16] with 10cm edge size. When using a sensor with a limited field of view (FoV), the beam based sensor model may lead to false free voxels on the edge of the FoV. Fig. 2 illustrates this edge case. As the beams on the edge of the FoV hit objects A an B, the voxels these beams intersect are erroneously marked as free. These false negatives should be minimized.

The original algorithm was modified to remove all free voxels that border unknown voxels. In the example given, this removes all the false negatives except(4,7). It also means that some voxels, such as (9,6) will be labelled as unknown instead of free. This trade-off is acceptable to remove the large numbers of false negatives.

In Fig. 2, there are also two beams which do not hit an obstacle and therefore have no points associated with them, causing the information about the free space they intersect to be lost. Pagad *et al.* [32] proposed the introduction of artificial endpoints. A virtual sphere of radius $r$ would be created and all voxels that the beam intersects between the camera position and the virtual sphere could be considered free. Applying this to the HoloLens 2, however, is not feasible as depth pixels without measurement may be also caused beams hitting reflective surfaces at shallow angles (such as glass or monitors) as well as discarded data due to multi-path interference, both common in indoors environments. Artificial endpoints might still be useful in outdoor scenes, however.

Moving object detection is achieved by comparing the occupancy state $S_t$ of each voxel observed in the current frame with its occupancy state $S_{t-1}$ from the previous frame. There are six possible cases which may occur for each voxel:

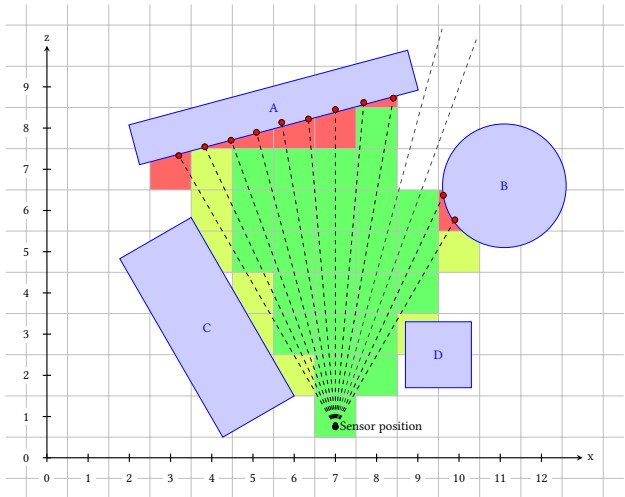

**Figure 2: A case in which voxels may falsely be classified as being free. Red voxels are occupied, green voxels are free, yellowish green voxels are misclassified as free and white voxels have an unknown occupancy.**

(1) $S_{t-1} = free$ and $S_t = occupied$: The voxel is added to a list of possible dynamic voxels.
(2) $S_{t-1} = occupied$ and $S_t = free$: If the voxel is observed as free over multiple consecutive frames, it is removed from the list of possible dynamic voxels.
(3) $S_{t-1} = S_t = occupied$: Voxel is part of the map of the static environment.
(4) $S_{t-1} = unobserved$ and $S_t = occupied$: Voxel assumed static and added to the map of the static environment.
(5) $S_{t-1} = S_t = free$: The voxel remains free in the map of the static environment
(6) $S_{t-1} = unobserved$ and $S_t = free$: The voxel is added as free space to the map of the static environment.

We implemented a check ensuring that all possible dynamic voxels have been observed recently. In case a possible dynamic voxel has not been observed in the last 15 frames, it is assumed to be no longer within the sensor's field of view and is therefore removed from the list of possible dynamic objects.

The ToF depth sensor of HoloLens 2 produces noisier measurements than a LiDAR sensor, which caused voxels directly above the floor to sometimes be detected as dynamic. As these voxels will cause issues when clustering dynamic voxels, they need to be detected and discarded. These voxels are detected by estimating the height of the floor and removing all dynamic voxels which are directly above the estimated floor height. For each frame, the height of the lowest occupied voxel is determined as the new potential floor height. If this new potential floor height is below the currently estimated floor height and there are more than 100 occupied voxels at that height, the currently estimated floor height is set to the new potential floor height.

The list of voxels possibly belonging to dynamic objects is then clustered using a region growing algorithm, with clusters not containing enough voxels being discarded.

In [4], each cluster is then classified into one of four classes - car, bicycle/motorbike, bus/truck and pedestrian. The classification criteria used are the size of the bounding box and the ratios between bounding box dimensions. This simple classification works in the autonomous driving scenario where classes have large size differences, however for most foreseeable MOTH scenarios such a classification is not applicable.

For tracking, the authors proposed using Global Nearest Neighbour (GNN) to search for neighbouring clusters in two different frames. If the clusters are different classes, they are discarded, otherwise they are added to the current track estimated with a Kalman filter. For track management, any observation which could not be associated with an already existing track is either noise or a candidate for a new track. Each unassociated observation results in the creation of a new track is marked as being not confirmed. Once a non-confirmed track gets associated to observations in the next frames, the track is marked as confirmed, otherwise it is discarded.

To summarise, the method described in [4] was implemented and modified to adapt to the HoloLens 2. We modified the method to filter out voxels which may be misclassified due to the limited FoV of the ToF sensor. We proved the use of artificial endpoints described in [32] to not be feasible in indoor environments. We implemented an additional check on dynamic voxels. Finally, we estimated the floor height to remove the noisy voxels present there.

## 6 CLUSTERING BASED APPROACH

The approach proposed in [46] consists of four components: a cluster detector which segments the point cloud observed by the sensor into multiple clusters; a human classifier which labels each cluster as being a human or a non-human; a multi-target tracker that tracks the clusters over time; a sample generator that can use the clusters, their corresponding labels and the tracking information to generate additional training data for retraining and fine-tuning the human classifier online. The code is also available on GitHub[3].

To detect the clusters, the ground plane needs to first be removed. In [46], all points below a certain z-value are removed. This works well for flat ground planes and a predictable starting height, such as in [46]. However, the origin of the HoloLens' coordinate system is its location at the start of the application. This means that the distance to the floor varies with user height and pose. We initalize the floor height at 0.0 and search for the largest plane roughly perpendicular to the z-axis using RANSAC.

Clusters are extracted using euclidean clustering with an adaptive cluster distance threshold $d^*$ calculated by Eq. 1 as points observed further from the sensor have a lower point density than points measured in close proximity to the sensor.

$$d^* = 2 \cdot r \cdot tan\frac{\theta}{2} \qquad (1)$$

where $\theta$ is the angular resolution of the sensor and $r$ the distance to the sensor.

Since having an adaptive distance for each point would be computationally intensive, Yan *et al.* [46] proposed to divide the point cloud into concentric cylindrical slices around the sensor. The number of slices and a fixed $\Delta d$ are selected. The $d^*$ of each slice, starting

---

[3]https://github.com/yzrobot/online_learning

from the one nearest to the sensor is then incremented by $\Delta d$. To calculate the radius of each slice, the floor of the inverse of Eq. 1 in regards to $r$ is calculated. Each slice is then clustered by the same adaptive threshold.

As a last step, clusters that are smaller or larger than a predetermined bounding box are removed from the detected clusters $C_j$, keeping only clusters that may realistically contain a human. More precisely, the retained clusters are all defined by $\overline{C} = \{C_j \mid 0.2 \leq w_j \leq 1, \ 0.2 \leq d_j \leq 1, \ 0.2 \leq h_j \leq 2\}$ where $w_j$ is the width, $d_j$ the depth and $h_j$ the height of the bounding box of cluster $C_j$.

Similar to the voxel based approach, the centroids of the detected clusters are then tracked using an UKF with a GNN data association scheme. The tracking is done in 2D, as it is assumed that the objects move parallel to the ground plane. The prediction model assumes constant object velocity between two frames.

Parallel to the tracking, a SVM classifies each cluster as either a human or a non-human object. The SVM can initially be trained with a small set of manually annotated samples. Using the tracking data from the cluster tracker and previous classification results from the SVM, misclassifications made by the SVM can be used to automatically generate more training samples. These additional training samples can then be used to retrain the SVM online.

In order to determine whether a cluster corresponds to a human, 7 different features with a total feature vector of 71 dimensions are extracted from each cluster. The list of features and their description can be found Table 1 in [46].

A binary SVM with a Gaussian Radial Basis Function kernel uses these features to calculate the probability of features corresponding to a human inside the cluster. We added a second SVM using the same features to track moving robot arms for human-robot collaboration. The last 25 classes assigned to the object are stored and the assigned class is the most often detected one.

Due to the tracking algorithm of [4] being almost identical to the tracking algorithm of [46], it was chosen to reuse the cluster tracker from this approach also for the voxel grid based approach, as it is more applicable to our use case.

To summarise, we implemented the method presented in [46]. We added a RANSAC-based estimation of the ground plane. We also added and trained a second SVM to track moving robot arms. We also modified the previous voxel based approach to include the tracking and classification pipeline proposed in [46].

## 7 EXPERIMENTAL EVALUATION

Two datasets were taken. The first dataset consists of a sequence of depth images with the corresponding self-localization information of the HoloLens 2 in which the HoloLens 2 was first moved around an indoor area for about one minute and 20 seconds. During this initial "mapping" part of the sequence, no moving object was visible to the HoloLens 2. After that, the HoloLens 2 was placed down on a table, and a human walked around the area for about one minute and 15 seconds, entering and leaving the depth sensor's field of view six times in total. The goal of this first dataset is to test the detection, classification and tracking of the humans.

In the second dataset an industrial robot arm was moving in cluttered spaces instead of a human.

## 7.1 Experiment 1: Qualitative evaluation of the voxel grid based approach

In the first dataset, some objects were falsely detected as being moving objects during the mapping step, mostly those with reflective surfaces. During the classification step, all of these false detections were correctly classified as background objects, thus not being tracked by the object tracker. The human was correctly detected and classified in most frames, with few frames where they were misclassified as background object. In one particular case, one leg was partially occluded behind the other, resulting in two clusters (Fig. fig. 3). Both clusters were classified as human which in turn led to a tracking error.

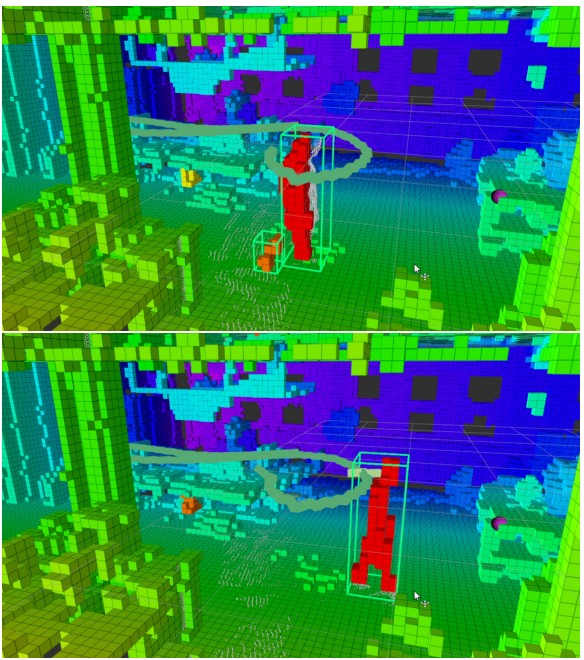

**Figure 3: Human Tracking errors of the voxel based approach - Top: Human is clustered into two clusters; Bottom: The resulting tracking error.**

Results were similar to the first dataset. There were some frames in which one monitor and part of a table were falsely classified as being a human, as can be seen by the green bounding box in fig. 4 (top). The misclasification was short enough that the object tracker did not track them. The robot was initially stationary, but as it moved around its work space, more and more parts of it were detected as being a moving object. Similar to the case in the first dataset, obstruction resulted in the robot being segmented into two clusters, one of which was incorrectly classified as being a human, as can be seen in fig. 4 (bottom). However, the clustering error did not result in a tracking error.

## 7.2 Experiment 2: Qualitative evaluation of the clustering based approach

Similar to the first approach, the human was detected correctly in most frames, with some misclassification as background object.

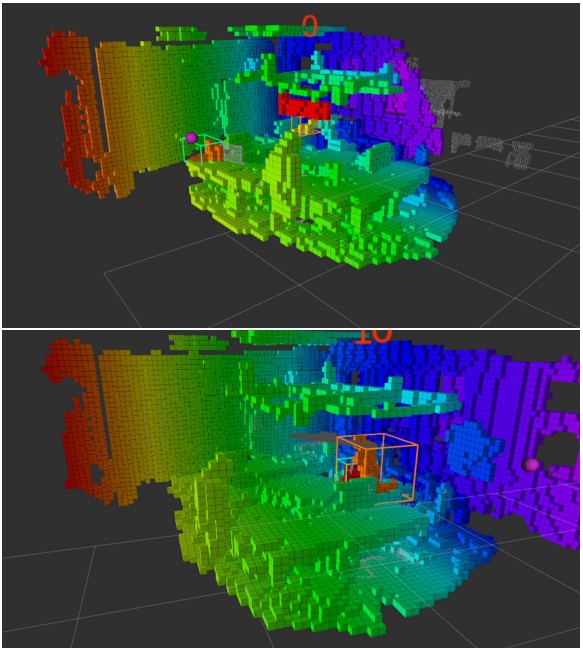

Figure 4: Robot Tracking errors of the voxel based approach - Top: part of monitor and table misclassified as human; Bottom: robot arm clustered into two clusters with one cluster being misclassified.

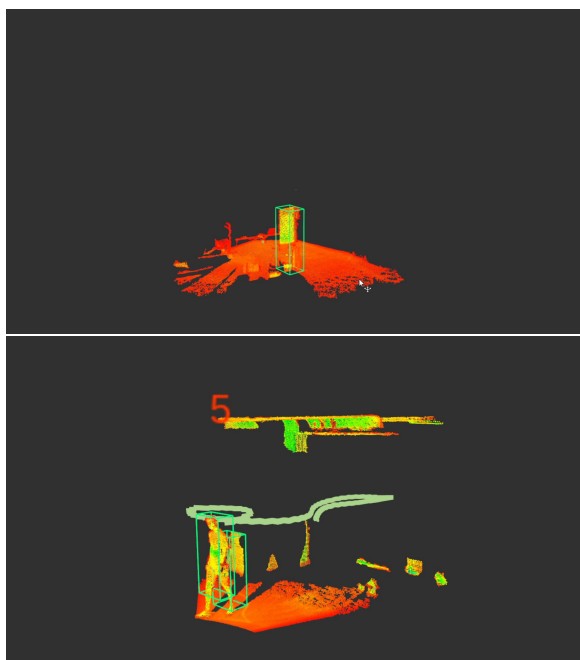

Figure 5: Human Tracking errors of the cluster based approach - Top: pillar misclassified as human. Bottom: human clustered into two clusters.

The clustering algorithm also sometimes segmented the human subject into two clusters, as can be seen in fig. 5 (top), but none of these clustering errors lead to tracking errors. There were some false detections however, e.g. a pillar supporting the ceiling was sometimes falsely classified as human when it was only partially visible (fig. 5 bottom).The misclassification lasted only a few frames, thus the object tracker did not start tracking them.

For the second dataset mixed results were observed. The robot arm was located in a cluttered space, therefore the clustering algorithm was not always able to correctly segment the point cloud into separate clusters. This resulted in the robot being sometimes segmented into multiple clusters, as can be seen in fig. 6 (top), or added to the same cluster as the surrounding objects, resulting in the robot not being detected. Due to the robot being detected so sporadically, the object tracking algorithm could not track it. The robot arm itself was also sometimes falsely classified as being a human, as can be seen in fig. 6 (bottom).

## 7.3 Experiment 3: Quantitative evaluation of the object tracking accuracy

To get a quantitative evaluation, an ARTrack2 IR tracking system was used to provide ground truth. The dataset captured for this experiment consists of the raw depth sensor data and the corresponding self-localization information, both obtained by the HoloLens 2, and the position of the tracking marker obtained by the ART tracking system. A person carrying an IR marker walked inside the tracking area for 97 seconds. To register the tracking data between

the ART tracker and the HoloLens 2, we used the first four points in which the person was detected as stationary (without significant movement of the marker). The person was tracked in 2D, meaning the height was disregarded.

The centroid of each MOTH detection was compared to the position of the tracking system's marker at the same time. This gives an estimate of the tracking error including all the communication and computing delays. Secondly, each trajectory obtained by MOTH was compared to the trajectory obtained by the tracking system using the CloudCompare software, which gives an estimate of the tracking error excluding delays. The results are given in Table 1.

Table 1: Tracking results of the two MOT approaches.

|  |  | Voxel Grid | | Clustering |
|  |  | map updates | no map updates |  |
| --- | --- | --- | --- | --- |
|  | avg (m) | 0.155 | 0.112 | 0.257 |
| Total error | max (m) | 0.578 | 0.331 | 0.773 |
|  | n detections | 455 | 472 | 177 |
|  | avg(m) | 0.008 | 0.008 | 0.008 |
| Registration error | max(m) | 0.014 | 0.014 | 0.015 |
|  | n detections | 4 | 4 | 4 |
|  | avg(m) | 0.021 | 0.020 | 0.023 |
| Stationary error | max(m) | 0.055 | 0.062 | 0.060 |
|  | n detections | 100 | 103 | 33 |
|  | avg(m) | 0.195 | 0.140 | 0.320 |
| Moving error | max(m) | 0.578 | 0.331 | 0.773 |
|  | n detections | 351 | 365 | 140 |
|  | mean(m) | 0.013 | 0.013 | 0.016 |
| Cloud to cloud distance | std. dev.(m) | 0.011 | 0.011 | 0.017 |
| (via CloudCompare) | n detections | 455 | 472 | 177 |
|  | n ref. points | 1104) | 1104 | 1104 |

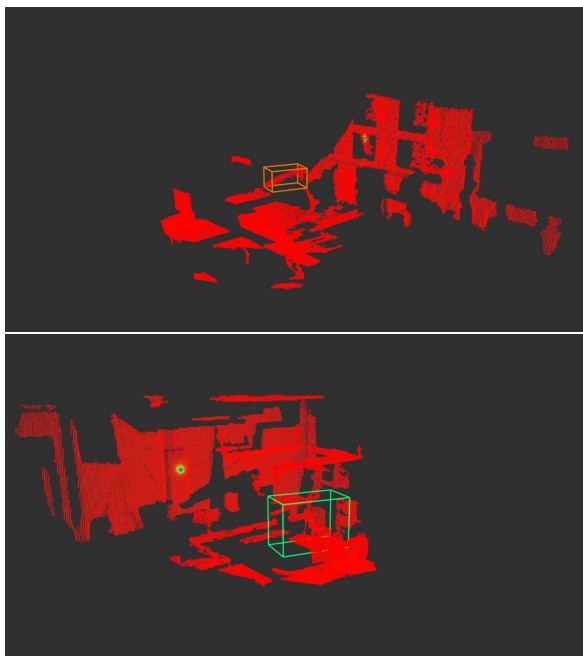

**Figure 6: Robot Tracking errors of the cluster based approach - Top: Robot arm clustered into two clusters; Bottom: robot arm misclassified as human**

The voxel grid based approach[4] was able to process more frames (455 and 472 frames in total with and without updates of the spatial map respectively) than the clustering based approach[45, 46] (177 frames in total). The clustering based approach had a total error of about 26 centimeters on average while the voxel grid based approach had an average total error of about 16 centimeters and 11 centimeters for the configuration with and without map updates respectively. When looking only at the object detections obtained while the external tracking system's marker was determined to be moving, this difference in tracking errors becomes even larger with the clustering based approach showing an average error of about 32 centimeters and the voxel grid based approach having average errors of about 19 and 14 centimeters for its two configurations respectively. While the person was stationary, the tracking error was 2 centimeters for all approaches. Similar results could be seen for the cloud to cloud distance calculated by CloudCompare, where both configurations of the voxel grid based approach achieved a mean distance of 1.3 centimeters, whereas the clustering based approach achieved an insignificantly worse mean distance of 1.6 centimeters. Both approaches have a similar tracking accuracy when ignoring processing delays, while the voxel grid based approach yields more accurate results otherwise.

## 8 DISCUSSION

The voxel grid based approach proved faster and slightly more precise, as well as working better in cluttered scenes. The clustering based approach is able to detect *movable* objects of interest even if these objects did not move at all, which has some use cases[13, 42].

Another point to discuss is the approaches' ability to create an additional map of the static environment. The voxel grid based approach has the benefit that it also creates a spatial map of the static environment. Depending on the application, this map may be useful for additional tasks such as robot cell setup and trajectory planning[36] or for background subtraction techniques. While the creation of such a map may be beneficial in some cases, it can also be a drawback in other cases. The clustering based approach can detect objects of interest with just a single frame given, whereas the voxel grid based approach first needs classify the voxels into a static map or as belonging to moving objects. If a moving object appears in a previously unmapped area, the voxel based approach will require several frames to detect the dynamic object, whereas the cluster based approach will require only one. In a static mapped environment there is no difference.

Our code is available for implementation and testing: The modified cloud annotation tool from [46] can be found at https://github.com/FabianB98/cloud_annotation_tool, the HoloLens 2 code at https://github.com/FabianB98/HoloLens2-ResearchMode-Unity and the ROS code at https://github.com/FabianB98/RosHololens2CatkinWs.

## 9 CONCLUSION AND FUTURE WORK

We implemented and tested two different approaches to MOTH - moving object tracking with HMDs. They were selected after a thorough review of MOT approaches, mostly in the fields of autonomous vehicles and mobile robots. The selected approaches met the criteria of being fast, computationally light and precise. These approaches were adapted from their original use cases to be efficiently used with HMDs. Tests have proven that both the voxel and clustering based approaches were feasible for tracking both humans and robots using an HMD in an indoor environment [4].

In regards to future work, we mentioned that the same approaches may be used in a variety of applications. As proof of concept, we already tested detection and tracking of a basketball [5]. Though the object classifier failed to identify the moving object as a basketball due to the lack of training data, the ball is clearly visible as a collection of dynamic voxels.

In the future we will further optimize the approaches and run it entirely on the HoloLens 2. Likewise, we would like to have more object categories, which can be tracked together or separately.

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
