# OpenReview forum: "MOTH: Moving Object Tracking via Head-Mounted Displays"
_humanrobotinteraction.org/HRI/2023/Workshop/VAM-HRI — VAM-HRI 2023 Oral_

### Official Review · Program_Chairs · 2023-02-24
**Accept**

**Rating:** 8
**Confidence:** 5

**Review:**

The paper describes the review of different methods for tracking moving objects using HoloLens 2. Authors go through different methods for tracking objects, providing detailed reviews and testing the methods feasible for their sensor and computational setup. Testing different publicly available methods is quite beneficial for the research from the practical point of view and can help other researchers to identify what methods to use for their experiments.

Potential improvements:

Check editing/spelling. A lot of minor mistakes, such as a lack of spaces, etc.

If the main contribution of the paper is testing different tracking methods (not modifying or improving them) I would expect more of them to be tested.

10cm resolution seems a bit too low for indoor object detection and tracking. That can be the reason for some of the objects not being detected or misclassified. It would be also interesting to see how different resolution impacts the results. Although experiment 2 seems to have a smaller grid size and also some misclassification

The results table can be improved to be more readable.

I see that there is a robotic arm present in the studies but I do not fully see the human-robot interaction component of this study. I think it is great practical work, but to make it more suitable for the HRI community, I would modify it a bit/add e.g. collaborative tasks where you try to track the objects.


-----------------------------------

Reviewer 2

This paper examines methods of moving object detection (MOD) and (moving object tracking) constrained to the hardware limitations found on modern HMDs. This paper provides a clear motivation and related work to the reader and is a strength of this submission. Two methods of MOT were selected for evaluation, the justification of which is clearly presented in Section 3. The implementation and evaluation provide the reader with a strong discussion regarding the strengths and weaknesses of the voxel grid and clustering approaches which provides the VAM-HRI community with valuable design guidance when implementing systems that have computational hardware limitations. This is well-written and a highly relevant paper for the VAM-HRI Workshop. I advocate for this submission’s acceptance.

Questions and Comments:
- Great figures that help the reader understand the MOT approaches as well as understand the discussion of results.
- To my knowledge, this is a very novel analysis that addresses the major limitations of VAM HMDs that are often disregarded in laboratory settings (through the use of motion tracking systems, or external computational hardware).
- Figure 6 could use additional explanation to better understand the content in the two images.

---

### Decision · Program_Chairs · 2023-03-02

Accept (Oral)